# `FedSplit`: an algorithmic framework for fast federated optimization

**Reese Pathak**
Department of Electrical Engineering and Computer Sciences
University of California, Berkeley
Berkeley, CA 94720
`pathakr@berkeley.edu`

**Martin J. Wainwright**
Department of Electrical Engineering and Computer Sciences
Department of Statistics
University of California, Berkeley
Berkeley, CA 94720
`wainwrig@berkeley.edu`

## Abstract

Motivated by federated learning, we consider the hub-and-spoke model of distributed optimization in which a central authority coordinates the computation of a solution among many agents while limiting communication. We first study some past procedures for federated optimization, and show that their fixed points need not correspond to stationary points of the original optimization problem, even in simple convex settings with deterministic updates. In order to remedy these issues, we introduce `FedSplit`, a class of algorithms based on operator splitting procedures for solving distributed convex minimization with additive structure. We prove that these procedures have the correct fixed points, corresponding to optima of the original optimization problem, and we characterize their convergence rates under different settings. Our theory shows that these methods are provably robust to inexact computation of intermediate local quantities. We complement our theory with some experiments that demonstrate the benefits of our methods in practice.

## 1 Introduction

Federated learning is a rapidly evolving application of distributed optimization for learning problems in large-scale networks of remote clients [13]. These systems present new challenges, as they are characterized by heterogeneity in computational resources, data across a large, multi-agent network, unreliable communication, and privacy constraints due to sensitive client data [15].

Although distributed optimization has a rich history and extensive literature (e.g., see the sources [2, 4, 8, 28, 14, 23] and references therein), renewed interest due to federated learning has led to a flurry of recent work in the area. Notably, McMahan et al. [17] introduced the `FedSGD` and `FedAvg` algorithms, by adapting the classical stochastic gradient method to the federated setting, considering the possibility that clients may fail and may only be subsampled on each round of computation. Another recent proposal, `FedProx`, attempted to mitigate potential device heterogeneity issues by applying averaged proximal updates to solve federated minimization problems. Currently, a general convergence theory of these methods is lacking. Moreover, practitioners have documented failures of convergence in certain settings (*e.g.*, see Figure 3 and related discussion in the work [17]).

**Our contributions:** The first contribution of this paper is to analyze some past procedures, and show that even in the favorable setting of deterministic updates (i.e., no stochastic approximation used), these methods typically fail to preserve solutions of the original optimization problem as fixed points. More precisely, even when these methods do converge, the resulting fixed point need *not* correspond to an optimal solution of the desired federated learning problem. Since the stochastic variants implemented in practice are approximate versions of the underlying deterministic procedures, this implies these methods also fail to preserve the correct fixed points in general.

With the motivation of rectifying this undesirable feature, our second contribution is to introduce a family of federated optimization algorithms, which we call `FedSplit`, that do preserve the correct fixed points for distributed optimization problems of the form

$$\text{minimize} \quad F(x) := \sum_{j=1}^{m} f_j(x), \tag{1}$$

where $f_j \colon \mathbf{R}^d \to \mathbf{R}$ are the clients' cost functions for variable $x \in \mathbf{R}^d$. In machine learning applications, the vector $x \in \mathbf{R}^d$ is a parameter of a statistical model. Our procedure and analysis builds on a long line of work relating optimization with monotone operators and operator splitting techniques [4, 26, 7, 1]. In this paper, we focus on the case when $f_j$ are convex functions with Lipschitz continuous gradient [24].

## 2 Existing algorithms and their fixed points

We focus our discussion on deterministic analogues of two recently proposed procedures—namely, `FedSGD` [17] and `FedProx` [16]. For analysis, it is useful to introduce the equivalent, consensus reformulation [4] of the distributed problem (1):

$$\begin{aligned} \text{minimize} \quad & F(x) := \sum_{j=1}^{m} f_j(x_j) \\ \text{subject to} \quad & x_1 = x_2 = \cdots = x_m. \end{aligned} \tag{2}$$

### 2.1 Federated gradient algorithms

The recently proposed `FedSGD` method [17] is based on a multi-step projected stochastic gradient method for solving the consensus problem. For our analysis we consider the obvious deterministic version of this algorithm, which replaces the stochastic gradient by the full gradient. Formally, given a stepsize $s > 0$, define the *gradient mappings*

$$G_j(x) := x - s\nabla f_j(x) \qquad \text{for } j = 1, \dots, m. \tag{3}$$

For a given integer $e \geqslant 1$, we define $G_j^e$ as the $e$-fold composition of $G_j$ and $G_j^0$ as the identity operator on $\mathbf{R}^d$. The `FedGD`$(s, e)$ algorithm from initialization $x^{(1)}$ obeys the recursion for $t = 1, 2, \dots$:

$$x_j^{(t+1/2)} := G_j^e(x_j^{(t)}), \qquad \text{for } j \in [m] := \{1, 2, \dots, m\}, \text{ and} \tag{4a}$$

$$x_j^{(t+1)} := \overline{x}^{(t+1/2)}, \qquad \text{for } j \in [m]. \tag{4b}$$

Recall that $\overline{x}^{(t+1/2)} = \frac{1}{m} \sum_{j=1}^{m} x_j^{(t+1/2)}$ is the block average. The following result characterizes the fixed points of this procedure.

**Proposition 1.** *For any $s > 0$ and $e \geqslant 1$, the sequence $\{x^{(t)}\}_{t=1}^{\infty}$ generated by the `FedGD`$(s, e)$ algorithm in equation (4) has the following properties: (a) if $x^{(t)}$ is convergent, then the local variables $x_j^{(t)}$ share a common limit $x^\star$ such that $x_j^{(t)} \to x^\star$ as $t \to \infty$ for $j \in [m]$; (b) any such limit $x^\star$ satisfies the fixed point relation*

$$\sum_{i=1}^{e} \sum_{j=1}^{m} \nabla f_j(G_j^{i-1}(x^\star)) = 0. \tag{5}$$

The proof of this claim, as well as all other claims in the paper, are deferred to Appendix A of the supplement.

Unpacking this claim slightly, suppose first that a single update is performed between communications, so $e = 1$. In this case, we have $\sum_{i=1}^{e} \nabla f_j(G_j^{i-1}(x^\star)) = \nabla f_j(x^\star)$, so that if $x^{(t)}$ has a limit $x$, it satisfies the relations

$$x_1 = x_2 = \cdots = x_m \quad \text{and} \quad \sum_{j=1}^{m} \nabla f_j(x_j) = 0.$$

Consequently, provided that the losses $f_j$ are convex, Proposition 1 implies that the limit of the sequence $x^{(t)}$, when it exists, is a minimizer of the consensus problem (2).

On the other hand, when $e > 1$, a limit of the iterate sequence $x^{(t)}$ must satisfy equation (5), which in general causes the method to have limit points which are not minimizers of the consensus problem. We give a concrete example in Section 2.3.

## 2.2 Federated proximal algorithms

Another recently proposed algorithm is `FedProx` [16], which can be seen as a distributed method loosely based on the classical proximal point method [24]. For a given stepsize $s > 0$, the *proximal operator* of a function $f: \mathbf{R}^d \to \mathbf{R}$ and its associated optimal value, the *Moreau envelope* of $f$, are given by [19, 24, 25, chap. 1.G]:

$$\mathbf{prox}_{sf}(z) := \arg\min_{x \in \mathbf{R}^d} \left\{ f(x) + \frac{1}{2s}\|z - x\|^2 \right\} \quad \text{and} \quad M_{sf}(z) := \inf_{x \in \mathbf{R}^d} \left\{ f(x) + \frac{1}{2s}\|z - x\|^2 \right\}.$$

We remark that when $f$ is convex, the existence of such a (unique) minimizer for the problem implied by the proximal operator is immediate.

With these definitions in place, we can now study the behavior of the `FedProx` method [16]. We again consider a deterministic version of `FedProx`, in which we remove any inaccuracies introduced by stochastic approximation. For a given initialization $x^{(1)}$, for $t = 1, 2, \ldots$:

$$x_j^{(t+1/2)} := \mathbf{prox}_{sf_j}(x_j^{(t)}), \qquad \text{for } j \in [m], \text{ and} \tag{6a}$$

$$x_j^{(t+1)} := \overline{x}^{(t+1/2)}, \qquad \text{for } j \in [m]. \tag{6b}$$

The following result characterizes the fixed points of this method.

**Proposition 2.** *For any stepsize $s > 0$, the sequence $\{x^{(t)}\}_{t=1}^{\infty}$ generated by the `FedProx` algorithm (see equations (6a) and (6b)) has the following properties: (a) if $x^{(t)}$ is convergent then, the local variables $x_j^{(t)}$ share a common limit $x^\star$ such that $x_j^{(t)} \to x^\star$ as $t \to \infty$ for each $j \in [m]$; (b) the limit $x^\star$ satisfies the fixed point relation*

$$\sum_{j=1}^{m} \nabla M_{sf_j}(x^\star) = 0. \tag{7}$$

Hence, we see that this algorithm has fixed points that will be a zero of the sum of the gradients of the Moreau envelopes $M_{sf_j}$, rather than a zero of the sum of the gradients of the functions $f_j$ themselves. When $m > 1$, these fixed point relations are, in general, different.

It is worth noting a very special case in which `FedGD` and `FedProx` will preserve the correct fixed points, even when $e > 1$. In particular, suppose *all* of local cost functions share a common minimizer $x^\star$, so that $\nabla f_j(x^\star) = 0$ for $j \in [m]$. Under this assumption, we have $G_j(x^\star) = x^\star$ all $j \in [m]$, and hence by arguing inductively, we have $G_j^i(x^\star) = x^\star$ for all $i \geqslant 1$. Additionally recall that the minimizers of $f_j$ and $M_{sf_j}$ coincide. Consequently, the fixed point relations (5) and (7) corresponding to `FedGD` and `FedProx` respecively, are both equivalent to the optimality condition for the federated problem. However, we emphasize this condition is *not realistic* in practice: if the optima of $f_j$ are exactly (or even approximately) the same, there would be little point in sharing data between devices by solving the federated learning problem. In contrast, the `FedSplit` algorithm presented in the next section retains correct fixed points for general federated learning problems without making such unrealistic, additional assumptions.

## 2.3 Example: Incorrectness on a least squares problem

We illustrate these non-convergence results by specializing to least squares and carrying out a simulation study on a synthetic least squares dataset. For $j = 1, \ldots, m$, suppose that we are given a design matrix $A_j \in \mathbf{R}^{n_j \times d}$ and a response vector $b_j \in \mathbf{R}^{n_j}$. The least squares regression problem defined by all the devices takes the form

$$\text{minimize} \quad F(x) := \frac{1}{2} \sum_{j=1}^{m} \|A_j x - b_j\|^2. \tag{8}$$

This problem is a special case of our general problem (1) with $f_j(x) = (1/2)\|A_j x - b_j\|^2$ for all $j$. When $A_j$ are full rank, the solution to this problem is unique and given by

$$x_{\mathrm{ls}}^\star = \left( \sum_{j=1}^{m} A_j^\mathsf{T} A_j \right)^{-1} \sum_{j=1}^{m} A_j^\mathsf{T} b_j. \tag{9}$$

Following Proposition 2, it is easy to verify that `FedProx` has fixed points of the form

$$x_{\mathtt{FedProx}}^\star = \left( \sum_{j=1}^{m} \left\{ I - (I + sA_j^\mathsf{T} A_j)^{-1} \right\} \right)^{-1} \left( \sum_{j=1}^{m} (A_j^\mathsf{T} A_j + (1/s)I)^{-1} A_j^\mathsf{T} b_j \right).$$

Following Proposition 1, it is easy to verify that `FedGD` has fixed points of the form[1]

$$x_{\mathtt{FedGD}}^\star = \left( \sum_{j=1}^{m} A_j^\mathsf{T} A_j \left\{ \sum_{k=0}^{e-1} (I - sA_j^\mathsf{T} A_j)^k \right\} \right)^{-1} \left( \sum_{j=1}^{m} \left\{ \sum_{k=0}^{e-1} (I - sA_j^\mathsf{T} A_j)^k \right\} A_j^\mathsf{T} b_j \right). \tag{10}$$

Therefore the previous three displays show that in general, when $m > 1$ and $e > 1$—that is, with more than one client, and more than one local update between communication rounds—we have $x_{\mathtt{FedProx}}^\star \neq x_{\mathrm{ls}}^\star$ and $x_{\mathtt{FedGD}}^\star \neq x_{\mathrm{ls}}^\star$. Therefore, we see that `FedProx` and `FedGD` do not have the correct fixed points, even with idealized deterministic updates.

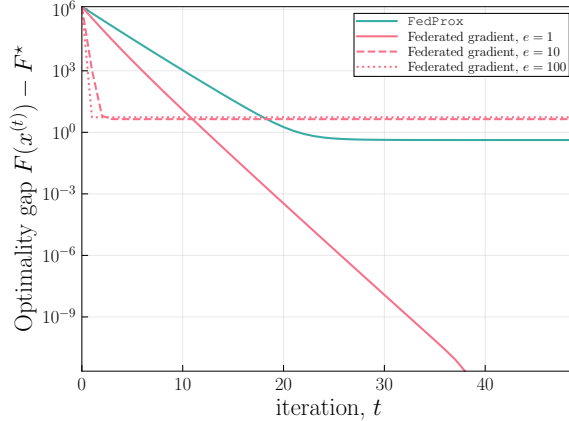

**Figure 1.** Plots of $F(x^{(t)}) - F^\star$ versus iteration number $t$ for a least-squares problem (8). This measures the difference between the optimal value and the value on round $t$ given by an algorithm.

Figure 1 shows the results of applying the (deterministic) versions of `FedProx` and `FedSGD`, with varying numbers of local epochs $e \in \{1, 10, 100\}$ for the least squares minimization problem (8). As expected, we see that `FedProx` and multi-step, deterministic `FedSGD` fail to converge to the correct fixed point for this problem. Although the presented deterministic variant of `FedSGD` will converge when a single local gradient step is taken between communication rounds (*i.e.*, when $e = 1$), we see that it also does not converge to the optimal solution as soon as $e > 1$. See Appendix B.1 of the supplement for additional details on this simulation study.

## 3  `FedSplit` and convergence guarantees

We now turn to the description of a framework that allows us to provide a clean characterization of the fixed points of iterative algorithms and to propose algorithms with convergence guarantees. Throughout our development, we assume that each function $f_j \colon \mathbf{R}^d \to \mathbf{R}$ is convex and differentiable.

### 3.1  An operator-theoretic view

We begin by recalling the consensus formulation (2) of the problem in terms of a block-partitioned vector $x = (x_1, \ldots, x_m) \in (\mathbf{R}^d)^m$, the function $F \colon (\mathbf{R}^d)^m \to \mathbf{R}$ given by $F(x) := \sum_{j=1}^m f_j(x_j)$, and the constraint set $E := \{x \mid x_1 = x_2 = \cdots = x_m\}$ is the feasible subspace for problem (2). By appealing to the first-order optimality conditions for the problem (2), it is equivalent to find a vector $x \in (\mathbf{R}^d)^m$ such that $\nabla F(x)$ belongs to the normal cone of the constraint set $E$, or equivalently such that $\nabla F(x) \in E^\perp$. Equivalently, if we define a set-valued operator $\mathcal{N}_E$ as

$$\mathcal{N}_E(x) := \begin{cases} E^\perp, & x_1 = x_2 = \cdots = x_m, \\ \emptyset, & \text{else} \end{cases} \tag{11}$$

then it is equivalent to find a vector $x \in (\mathbf{R}^d)^m$ that satisfies the inclusion condition

$$0 \in \nabla F(x) + \mathcal{N}_E(x). \tag{12}$$

where $\nabla F(x) = (\nabla f_1(x_1), \ldots, \nabla f_m(x_m))$.

When the loss functions $f_j \colon \mathbf{R}^d \to \mathbf{R}$ are convex, both $\nabla F$ and $\mathcal{N}_E$ are monotone operators on $(\mathbf{R}^d)^m$ [1]. Thus, the display (12) is a *monotone inclusion problem*. Methods for solving monotone inclusions have a long history of study within the applied mathematics and optimization literatures [26, 7]. We now use this framework to develop and analyze algorithms for solving the federated problems of interest.

### 3.2  Splitting procedures for federated optimization

We now describe a method, derived from splitting the inclusion relation, whose fixed points do correspond with global minima of the distributed problem. It is an instantiation of the Peaceman-Rachford splitting [20], which we refer to as the `FedSplit` algorithm in this distributed setting.

---

**Algorithm 1** [`FedSplit`] *Splitting scheme for solving federated problems of the form* (1)

**Given** initialization $x \in \mathbf{R}^d$, proximal solvers $\texttt{prox\_update}_j \colon \mathbf{R}^d \to \mathbf{R}^d$

**Initialize** $x^{(1)} = z_1^{(1)} = \cdots = z_m^{(1)} = x$

**for** $t = 1, 2, \ldots$:

    1. **for** $j = 1, \ldots, m$:

        a. *Local prox step:* set $z_j^{(t+1/2)} = \texttt{prox\_update}_j(2x^{(t)} - z_j^{(t)})$

        b. *Local centering step:* set $z_j^{(t+1)} = z_j^{(t)} + 2(z_j^{(t+1/2)} - x^{(t)})$

    **end for**

    2. *Compute global average:* set $x^{(t+1)} = \overline{z}^{(t+1)}$.

**end for**

---

Thus, the `FedSplit` procedure maintains a parameter vector $z_j^{(t)} \in \mathbf{R}^d$ for each device $j \in [m]$. The central server maintains a parameter vector $x^{(t)} \in \mathbf{R}^d$, which collects averages of the parameter estimates at each machine. The local update at device $j$ is defined in terms of a proximal solver $\texttt{prox\_update}_j(\cdot)$, which typically be approximate proximal updates $\texttt{prox\_update}_j(x) \approx \mathbf{prox}_{sf_j}(x)$, uniformly in $x \in \mathbf{R}^d$ for a suitable stepsize $s > 0$. We make the sense of this approximation precise when we state our convergence results in Section 3.3. An advantage to `FedSplit` is that unlike `FedGD` and `FedProx`, it has the correct fixed points for the distributed problem.

**Proposition 3.** *Suppose for some $s > 0$, $\texttt{prox\_update}_j(\cdot) = \mathbf{prox}_{sf_j}(\cdot)$, for all $j$. Suppose that $z^\star = (z_1^\star, \ldots, z_m^\star)$ is a fixed point for the* `FedSplit` *procedure, meaning that*

$$z_j^\star = z_j^\star + 2\left(\mathbf{prox}_{sf_j}(2\overline{z^\star} - z_j^\star) - \overline{z^\star}\right), \qquad \text{for all } j \in [m]. \tag{13}$$

*Then the average $x^\star := \frac{1}{m}\sum_{j=1}^m z_j^\star$ is optimal: $\sum_{j=1}^m f_j(x^\star) = \inf_{x \in \mathbf{R}^d} \sum_{j=1}^m f_j(x)$.*

### 3.3 Convergence results

In this section, we give convergence guarantees for the `FedSplit` procedure in Algorithm 1 under exact and inexact proximal operator implementations.

**Strongly convex and smooth losses**   We begin by considering the case when the losses $f_j \colon \mathbf{R}^d \to \mathbf{R}$ are $\ell_j$-strongly convex and $L_j$-smooth. We define

$$\ell_* := \min_{j=1,\dots,m} \ell_j, \quad L^* := \max_{j=1,\dots,m} L_j, \quad \text{and} \quad \kappa := \frac{L^*}{\ell_*}. \tag{14}$$

Note that $\kappa$ corresponds to the induced condition number of our federated problem (2).

The following result demonstrates that in this setting, our method enjoys geometric convergence to the optimum, even with inexact proximal implementations.

**Theorem 1.** *Consider the* `FedSplit` *algorithm with possibly inexact proximal implementations,*

$$\|\texttt{prox\_update}_j(z) - \mathbf{prox}_{sf_j}(z)\| \leqslant b \quad \text{for all } j \text{ and all } z \in \mathbf{R}^d, \tag{15}$$

*and with stepsize* $s = 1/\sqrt{\ell_* L^*}$. *Then for any initialization, the iterates satisfy*

$$\|x^{(t+1)} - x^\star\| \leqslant \left(1 - \frac{2}{\sqrt{\kappa}+1}\right)^t \frac{\|z^{(1)} - z^\star\|}{\sqrt{m}} + (\sqrt{\kappa}+1)b, \quad \text{for all } t = 1, 2, \dots. \tag{16}$$

We now discuss some aspects of Theorem 1.

**Exact proximal evaluations:**   In the special (albeit unrealistic) case when the proximal evaluations are exact, the uniform bound (15) holds with $b = 0$. Consequently, given some initialization $z^{(1)}$, if we want $\varepsilon$-accuracy, meaning $\|x^{(T)} - x^\star\| \leqslant \varepsilon$, we see that this occurs as soon as $T$ exceeds

$$T(\varepsilon, \kappa) = O(1) \left\{ \sqrt{\kappa} \log\left(\frac{\|z^{(1)} - z^\star\|}{\varepsilon\sqrt{m}}\right) \right\}$$

iterations of the overall procedure. Here $O(1)$ denotes a universal constant.

**Approximate proximal updates by gradient steps:**   In practice, the `FedSplit` algorithm will be implemented using an approximate prox-solver. Recall that the proximal update at device $j$ at round $t$ takes the form:

$$\mathbf{prox}_{sf_j}(x_j^{(t)}) = \underset{u \in \mathbf{R}^d}{\arg\min} \big\{ \underbrace{sf_j(u) + \frac{1}{2}\|u - x_j^{(t)}\|_2^2}_{h_j(u)} \big\}.$$

A natural way to compute an approximate minimizer is to run $e$ rounds of gradient descent on the function $h_j$. Concretely, at round $t$, we initialize the gradient method with the initial point $u^{(1)} = x_j^{(t)}$, and run gradient descent on $h_j$ with a stepsize $\alpha$, thereby generating the sequence

$$u^{(t+1)} = u^{(t)} - \alpha \nabla h_j(u^{(t)}) = u^{(t)} - \alpha s \nabla f_j(u^{(t)}) + \big(u^{(t)} - x_j^{(t)}\big) \tag{17}$$

We define $\texttt{prox\_update}_j(x_j^{(t)})$ to be the output of this procedure after $e$ steps.

**Corollary 1** (`FedSplit` convergence with inexact proximal updates)**.** *Consider the* `FedSplit` *procedure run with proximal stepsize* $s = \frac{1}{\sqrt{\ell_* L^*}}$, *and using approximate proximal updates based on $e$ rounds of gradient descent with stepsize* $\alpha = (1 + s\frac{\ell_* + L^*}{2})^{-1}$ *initialized (in round $t$) at the previous iterate* $x_j^{(t)}$. *Then the the bound (15) holds at round $t$ with error at most*

$$b \leqslant \big(1 - \frac{1}{\sqrt{\kappa}+1}\big)^e \|x_j^{(t)} - \mathbf{prox}_{sf_j}(x_j^{(t)})\|_2. \tag{18}$$

Given the exponential decay in the number of rounds $e$ exhibited in the bound (18), in practice, it suffices to take a relatively small number of gradient steps. For instance, in our experiments to be reported in Section 4, we find that $e = 10$ suffices to match the exact proximal updates. This inexact proximal update could also be implemented with a gradient method and backtracking line search [5].

**Smooth but not strongly convex losses** We now consider the case when $f_j \colon \mathbf{R}^d \to \mathbf{R}$ are $L_j$-smooth and convex, but not necessarily strongly convex. In this case, the consensus objective $F(z) = \sum_{j=1}^m f_j(z_j)$ is an $L^*$-smooth function on the product space $(\mathbf{R}^d)^m$.[2]

Our approach to solving such a problem is to apply the `FedSplit` procedure to a suitably regularized version of the original problem. More precisely, given some initial vector $x^{(1)} \in \mathbf{R}^d$ and regularization parameter $\lambda > 0$, let us define the function

$$F_\lambda(z) := \sum_{j=1}^m \left\{ f_j(z_j) + \frac{\lambda}{2m} \|z_j - x^{(1)}\|^2 \right\}. \tag{19}$$

We see that $F_\lambda \colon (\mathbf{R}^d)^m \to \mathbf{R}$ is a $\lambda$-strongly convex and $L_\lambda^* = (L^* + \lambda)$-smooth function. The next result shows that for any $\varepsilon > 0$, minimizing the function $F_\lambda$ up to an error of order $\varepsilon$, using a carefully chosen $\lambda$, yields an $\varepsilon$-cost-suboptimal minimizer of the original objective function $F$.

**Theorem 2.** *Given some $\lambda \in \left(0, \frac{\varepsilon}{m\|x^{(1)} - x^\star\|^2}\right)$ and any initialization $x^{(1)} \in \mathbf{R}^d$, suppose that we run the `FedSplit` procedure (Algorithm 1) on the regularized objective $F_\lambda$ using exact prox steps with stepsize $s = 1/\sqrt{\lambda L_\lambda^*}$. Then the `FedSplit` algorithm outputs a vector $\widehat{x} \in \mathbf{R}^d$ satisfying $F(\widehat{x}) - F^\star \leqslant \varepsilon$ after exceeding $\widetilde{O}\left(\sqrt{\frac{L^*\|x^{(1)} - x^\star\|^2}{\varepsilon}}\right)$ iterations.*[3]

We remark that this faster convergence rate of $\widetilde{O}\left(t^{-2}\right)$ is nearly optimal for first-order algorithms [18], and to our knowledge such results were not known for operator splitting-based procedures prior to this work.

## 4  Experiments

In this section, we present numerical results for `FedSplit` on some convex federated optimization problem instances. We include additional details on these simulations in Section B of the supplement.

**Logistic regression** We begin with federated binary classification, where we solve,

$$\text{minimize} \quad \sum_{j=1}^m \sum_{i=1}^{n_j} \log(1 + \mathrm{e}^{-b_{ij} a_{ij}^\top x}), \tag{20}$$

with variable $x \in \mathbf{R}^d$. We generate the problem data $\{(a_{ij}, b_{ij})\} \subset \mathbf{R}^d \times \{\pm 1\}$ synthetically; see Section B.2.1 in the supplement for details.

We also use `FedSplit` to solve a multiclass classification problem, with $K$ classes. Here we solve

$$\text{minimize} \quad \sum_{j=1}^m \left\{ \sum_{i=1}^{n_j} \sum_{k=1}^K \log(1 + \mathrm{e}^{-b_{ijk} a_{ij}^\top x_k}) + \frac{\lambda}{2} \sum_{k=1}^K \|x_k\|^2 \right\} \tag{21}$$

with variables $x_1, x_2, \ldots, x_K \in \mathbf{R}^d$, regularization parameter $\lambda > 0$, and sample size $N = \sum_{j=1}^m n_j$. Here, the problem data $\{(a_{ij}, b_{ij})\} \subset \mathbf{R}^d \times \{\pm 1\}^K$ are images and multiclass labels from the FEMNIST dataset in the LEAF framework [6]. This dataset was proposed as a benchmark for federated optimization; there are $N = 805{,}263$ images, $m = 3{,}550$ clients, and $K = 62$ classes. The problem dimension is $d = 6{,}875$; see Section B.2.2 in the supplement for additional details.

In Figure 2, we present numerical results on problems (20) and (21). We implement `FedSplit` with exact proximal operators and inexact implementations with a constant number of gradient steps $e \in \{1, 5, 10\}$. For comparison, we implemented a federated gradient method as previously described (4). As shown in Figure 2(a), both `FedGD` with $e = 1$ and the `FedSplit` procedure exhibit linear convergence rates. Using inexact proximal updates with the `FedSplit` procedure preserves the linear convergence up to the error floor introduced by the exactness of the updates. In this case, the

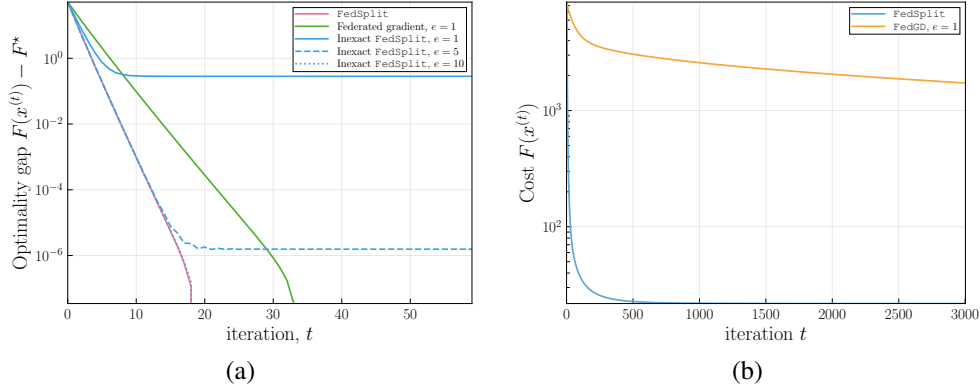

**Figure 2.** Cost versus iteration for `FedGD` and `FedSplit`. (a) Plot of the optimality gap $F(x^{(t)}) - F^\star$ versus the iteration number $t$ for the logistic regression problem (20). This measures the difference between the optimal cost value and cost of an iterate returned at round $t$ by a given algorithm. (b) Plot of cost $F(x^{(t)})$ versus iteration $t$ for the FEMNIST multiclass logistic problem (21).

inexact proximal updates with $e = 10$—that is, performing 10 local updates per each round of global communication—suffice to track the exact `FedSplit` procedure up to an accuracy below $10^{-6}$. In Figure 2(b), we see that `FedSplit` similarly outperforms `FedGD` on actual client data.[4]

**Dependence on problem conditioning**    It is well-known that the convergence rates of first-order methods are affected by problem conditioning. First, let us re-state our theoretical guarantees in terms of *iteration complexity*. We let $T(\varepsilon, \kappa)$ denote the maximum number of iterations required so that, for any problem with condition number at most $\kappa$, the iterate $x^{(T)}$ with $T = T(\varepsilon, \kappa)$ satisfies the bound $F(x^{(T)}) - F^\star \leqslant \varepsilon$. For federated objectives with condition number $\kappa$ as defined in (14), `FedSplit` and `FedGD` have iteration complexities

$$T_{\mathrm{FedSplit}}(\varepsilon, \kappa) = O(\sqrt{\kappa}\log(1/\varepsilon)) \quad \text{and} \quad T_{\mathrm{FedGrad}}(\varepsilon, \kappa) = O(\kappa\log(1/\varepsilon)). \tag{22}$$

This follows from Theorem 1 and standard results from convex optimization theory [18]. Hence, whereas `FedSplit` has a more expensive local update, it has much better dependence on the condition number $\kappa$. In the context of federated optimization, this iteration complexity should be interpreted as the number of communication rounds between clients and the coordinating entity. Hence, this highlights a concrete tradeoff between local computation and global communication in these methods. Note that while acclereated first-order methods matches the iteration complexity of `FedSplit`, they are sensitive to stepsize misspecification and are not robust to errors incurred in gradient updates [9]. This is in contrast to the inexact convergence guarantees that `FedSplit` enjoys (see Theorem 1).

In Figure 3, we present the results of a simulation study that shows these iteration complexity estimates are accurate in practice. We construct a sequence of least squares problems with varying condition number between 10 and 10000. We then look at the number of iterations required to obtain an $\varepsilon$-cost suboptimal solution with $\varepsilon = 10^{-3}$; see Section B.3 in the supplement for additional simulation details. In this way, we obtain estimates of the functions $\kappa \mapsto T_{\mathrm{FedGrad}}(10^{-3}, \kappa)$ and $\kappa \mapsto T_{\mathrm{FedSplit}}(10^{-3}, \kappa)$, which measure the dependence of the iteration complexity on the condition number. Figure 3 provides plots of these estimated functions.

Consistent with our theory, we see that `FedGD` has an approximately linear dependence on the condition number, whereas the `FedSplit` procedure has much milder dependence on conditioning. Concretely, for an instance with condition number $\kappa = 10000$, the `FedGD` procedure requires on the order of 34000 iterations, whereas the `FedSplit` procedure requires roughly 400 iterations. Therefore, while `FedSplit` involves more expensive intermediate proximal updates, it enjoys a smaller iteration count, which in the context of this federated setting indicates a significantly smaller number of communication rounds between clients and the the centralized server.

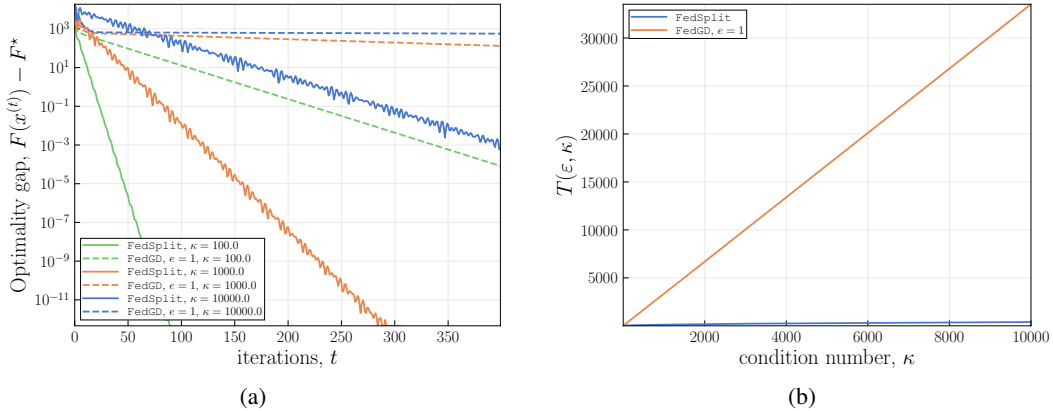

**Figure 3.** Dependence of algorithms on the conditioning. (a) Plot of log cost suboptimality of iterate $x^{(t)}$ versus iteration $t$ for condition number $\kappa \in \{100, 1000, 10000\}$. (b) Plots of the iteration complexity $T(\varepsilon; \kappa)$ versus $\kappa$ at tolerance level $\varepsilon = 10^{-3}$ for the `FedGD` and `FedSplit` procedures.

## 5 Discussion

We highlight a few interesting directions for future work on federated learning and `FedSplit`. First, in practice, it is standard to use stochastic optimization algorithms in solving large-scale machine learning problems, and we are currently analyzing stochastic approximation procedures as applied to the device-based proximal updates underlying our method. Our results on the incorrectness of previously proposed methods and the work of Woodworth and colleagues [27] on the suboptimality on multi-step stochastic gradient methods, highlight the need for better understanding of the tradeoff between the accuracy of stochastic and deterministic approximations to intermediate quantities and rates of convergence in federated optimization. We also mention the possibility of employing stochastic approximation with higher-order methods, such as the Newton sketch algorithm [21, 22]. It is also important to consider our procedure under asynchronous updates, perhaps under delays in computation. Finally, an important desideratum in federated learning is suitable privacy guarantees for client the local data [3]. Understanding how noise aggregated through differentially private mechanisms couples with our inexact convergence guarantees is a key direction for future work.

## Broader Impact

As mentioned in the introduction, a main application of federated optimization is to large-scale statistical learning, as carried out by application developers and cell phone manufacturers. On the other hand, learning from federated data is also inherent to other settings where data is not stored centrally: consider, for example, collecting clinical trial data across multiple hospitals and running a centralized analysis. Therefore, we envision analysts who are operating in these settings—where data is not available centrally due to communication barriers or privacy constraints—as main benefactors of this work. Our methods enjoy the same trade-offs with respect to biases in data, failures of systems, as other standard first-order algorithms. We believe that having convergent algorithms in this federated setting should help promote good practices with regard to analyzing large-scale, federated, and sensitive datasets.

## Acknowledgments and Disclosure of Funding

We thank Bora Nikolic and Cong Ma for their careful reading and comments of an initial draft of this manuscript. RP was partially supported by a Berkeley Fellowship via the ARCS Foundation. MJW was partially supported by Office of Naval Research grant DOD-ONR-N00014-18-1-2640, and NSF grant NSF-DMS-1612948.

## Footnotes

[1]Here we assume that $s > 0$ is small enough so that $\|I - sA_j^\mathsf{T} A_j\|_{\mathrm{op}} < 1$, which ensures convergence.

[2]To avoid degeneracies, we assume $x \mapsto \sum_{j=1}^m f_j(x)$ is bounded below and attains its minimum.

[3]The $\widetilde{O}\left(\cdot\right)$ notation denotes constant and polylogarithmic factors that are not dominant.

[4]Given the large scale nature of this example, we implement an accelerated gradient method for the proximal updates, terminated when the gradient of the proximal objective drops below $10^{-8}$.

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
