[Supplementary Material]



# Supplement to "`FedSplit`: an algorithmic framework for fast federated optimization"

## A   Proofs

We now turn to the proofs of our main results. Prior to diving into these arguments, we first introduce two operators that play a critical role in our analysis. Given a convex function $\varphi \colon \mathbf{R}^d \to \mathbf{R}$, we define

$$\mathbf{prox}_\varphi(z) := \arg\min_{x \in \mathbf{R}^d} \left\{ \varphi(x) + \frac{1}{2}\|z - x\|^2 \right\} \quad \text{and} \tag{21a}$$

$$\mathbf{refl}_\varphi(z) := 2\,\mathbf{prox}_\varphi(z) - z. \tag{21b}$$

These are called the proximal and reflected resolvent operators associated with the function $\varphi$. The first operator is also known as the resolvent; the second operator above is also known as the Cayley operator of $\varphi$. Moreover, our analysis makes use of the (semi)norm on Lipschitz continuous functions $f \colon \mathbf{R}^d \to \mathbf{R}$ given by

$$\mathrm{Lip}(f) := \sup_{x \neq y} \frac{|f(x) - f(y)|}{\|x - y\|}. \tag{22}$$

For short, we say that that $f$ is $\mathrm{Lip}(f)$-Lipschitz continuous when it satisfies this condition.

### A.1   Proofs of guarantees for `FedSplit`

We begin by proving our guarantees for the `FedSplit` procedure, including the correctness of its fixed points (Proposition 3); the general convergence guarantee in the strongly convex case (Theorem 1); the general convergence guarantee in the weakly convex case (Theorem 2), and Corollary 1 on its convergence with approximate proximal updates.

### A.2   Proof of Proposition 3

By the fixed point assumption, the block average $x^\star := \overline{z^\star}$ satisfies the relation

$$\mathbf{prox}_{sf_j}(2x^\star - z_j^\star) = x^\star \qquad \text{for } j = 1, 2, \ldots, m.$$

Since each $f_j$ is convex and differentiable, by the first-order stationary conditions implied by the definition of the prox operator (21a), we must have

$$\nabla f_j(x^\star) + \frac{1}{s}\left\{x^\star - \left(2x^\star - z_j^\star\right)\right\} \ = \ \nabla f_j(x^\star) + \frac{1}{s}\left\{z_j^\star - x^\star\right\} \ = \ 0 \quad \text{for } j = 1, \ldots, m.$$

Summing these equality relations over $j = 1, \ldots, m$ and using the fact that $x^\star = \frac{1}{m}\sum_{j=1}^m z_j^\star$ yields the zero gradient condition

$$\sum_{j=1}^m \nabla f_j(x^\star) \ = \ 0.$$

Since the function $x \mapsto \sum_{j=1}^m f_j(x)$ is convex, this zero-gradient condition implies that $x^\star \in \mathbf{R}^d$ is a minimizer of the distributed problem as claimed.

#### A.2.1   Proof of Theorem 1

We now turn to the proof of Theorem 1. Our strategy is to prove it as a consequence of a somewhat more general result, which we begin by stating here. In order to lighten notation, we use the fact that the proximal operator for the function $F(z_1, \ldots, z_m) = \sum_{j=1}^m f_j(z_j)$ is block-separable, so that in terms of the block-partitioned vector $z = (z_1, \ldots, z_m)$, we can write

$$\mathbf{prox}_{sF}(z) = \left(\mathbf{prox}_{sf_1}(z_1), \ldots, \mathbf{prox}_{sf_m}(z_m)\right), \quad \text{for all } z = (z_1, \ldots, z_m) \in (\mathbf{R}^d)^m.$$

We also recall the the approximate proximal operator used in the `FedSplit` procedure, namely

$$\widetilde{\mathbf{prox}}(z) := \left(\texttt{prox\_update}_1(z_1), \ldots, \texttt{prox\_update}_m(z_m)\right), \quad \text{for all } z_1, \ldots, z_m \in \mathbf{R}^d.$$

**Theorem 3** (Convergence with general residuals)**.** *Suppose that the functions $f_j \colon \mathbf{R}^d \to \mathbf{R}$ are $\ell_j$-strongly convex and $L_j$-smooth for $j = 1, \ldots, m$, and for $t = 1, 2, \ldots$, define the residuals*

$$r^{(t)} := \widetilde{\mathbf{prox}}(2\overline{z^{(t)}} - z^{(t)}) - \mathbf{prox}_{sF}(2\overline{z^{(t)}} - z^{(t)}). \tag{23}$$

*Then with stepsize $s = 1/\sqrt{\ell_* L^*}$, the* `FedSplit` *procedure (Algorithm 1) has a unique fixed point $z^\star$, and the iterates satisfy*

$$\|z^{(t+1)} - z^\star\| \leqslant \rho^t \|z^{(1)} - z^\star\| + 2 \sum_{j=1}^{t} \rho^{t-j} \|r^{(j)}\| \qquad \textit{for } t = 1, 2, \ldots, \tag{24}$$

*where $\rho := 1 - 2/(\sqrt{\kappa} + 1)$ is the contraction coefficient.*

Let us use Theorem 3 to derive the claim stated in Theorem 1. Note that by Proposition 3, the fixed points of Algorithm 1 are minimizers of $F$, hence unique under the strong convexity assumption. Consequently, we have

$$\|x^{(t+1)} - x^\star\| \leqslant \frac{1}{\sqrt{m}} \|z^{(t+1)} - z^\star\|, \qquad \text{for all } t = 1, 2, \ldots.$$

Using Theorem 3 and the error bound, we then conclude that

$$\|x^{(t+1)} - x^\star\| \leqslant \frac{1}{\sqrt{m}} \left(1 - \frac{2}{\sqrt{\kappa} + 1}\right)^t \|z^{(1)} - z^\star\| + (\sqrt{\kappa} + 1)b,$$

as claimed.

### A.2.2 Proof of Theorem 3

We now turn to the proof of the more general claim. Given additive decomposition $F(z) = \sum_{j=1}^{m} f_j(z_j)$, the reflected resolvent induced by $F$ is block-separable, taking the form

$$\mathbf{refl}_{sF}(z) = \left(\mathbf{refl}_{sf_1}(z_1), \ldots, \mathbf{refl}_{sf_m}(z_m)\right), \quad \text{for all } z = (z_1, \ldots, z_m) \in (\mathbf{R}^d)^m.$$

Similarly, consider the approximate reflected resolvent defined by the algorithm, namely

$$\widetilde{\mathbf{refl}}(z) := 2\widetilde{\mathbf{prox}}(z) - z, \quad \text{for all } z = (z_1, \ldots, z_m) \in (\mathbf{R}^d)^m.$$

It also has the same block-separable form.

Using these two block-separable operators, we can now define two abstract operators, each acting on the product space $(\mathbf{R}^d)^m$, that allow us to analyze the algorithm. The first operator $\mathcal{T}$ underlies the idealized algorithm, in which the proximal updates are exact, and the second operator $\widehat{\mathcal{T}}$ underlies the practical algorithm, which is based on approximate proximal updates. The idealized algorithm is based on iterating the operator

$$\mathcal{T}(z) := \mathbf{refl}_{sF}\left(\mathbf{refl}_{I_E}(z)\right). \tag{25}$$

In this definition, we use $I_E$ to denote the indicator function for membership in the equality subspace $E$, so that $\mathbf{refl}_{I_E}$ is the reflected proximal operator for this function.

On the other hand, the practical algorithm generates the sequence $\{z^{(t)}\}_{t=1}^{\infty}$ via the updates $z^{(t+1)} = \widehat{\mathcal{T}}(z^{(t)})$, where $\widehat{\mathcal{T}} \colon (\mathbf{R}^d)^m \to (\mathbf{R}^d)^m$ is the *perturbed operator*

$$\widehat{\mathcal{T}}(z) = \widetilde{\mathbf{refl}}\left(\mathbf{refl}_{I_E}(z)\right). \tag{26}$$

Note that the idealized operator $\mathcal{T}$ and perturbed operator $\widehat{\mathcal{T}}$ satisfy the relation

$$\widehat{\mathcal{T}} - \mathcal{T} = \left(\widetilde{\mathbf{refl}} \circ \mathbf{refl}_{I_E} - \mathbf{refl}_{sF} \circ \mathbf{refl}_{I_E}\right). \tag{27}$$

Our proof involves verifying that with the stepsize choice $s = 1/\sqrt{\ell_* L^*}$, the mapping $\mathcal{T}$ is a contraction, with Lipschitz coefficient

$$\mathrm{Lip}(\mathcal{T}) \leqslant \underbrace{1 - \frac{2}{\sqrt{\kappa} + 1}}_{=: \rho} < 1. \tag{28}$$

Taking this claim as given for the moment, the contractivity implies that $\mathcal{T}$ has has a unique fixed point [12]—call it $z^\star \in (\mathbf{R}^d)^m$. Comparing with Proposition 3, we see that the definition of fixed points given there agrees with the fixed point $z^\star$ of the operator $\mathcal{T}$, since we have the relation $\mathbf{refl}_{I_E}(z) = 2\bar{z} - z$.

Using this contractivity condition, the distance between this fixed point $z^\star$ and the iterates $z^{(t)}$ of the FedSplit procedure can be bounded as

$$
\begin{aligned}
\|z^{(t+1)} - z^\star\| &= \|\widehat{\mathcal{T}}z^{(t)} - \mathcal{T}z^\star\| \\
&\overset{\text{(i)}}{\leqslant} \|\mathcal{T}z^{(t)} - \mathcal{T}z^\star\| + 2\|\widehat{\mathbf{prox}}\,\mathbf{refl}_{I_E}\,z^{(t)} - \mathbf{prox}_{sF}\,\mathbf{refl}_{I_E}\,z^{(t)}\| \\
&\overset{\text{(ii)}}{\leqslant} \mathrm{Lip}(\mathcal{T})\|z^{(t)} - z^\star\| + 2\|r^{(t)}\| \\
&\overset{\text{(iii)}}{\leqslant} \rho\|z^{(t)} - z^\star\| + 2\|r^{(t)}\|,
\end{aligned}
\tag{29}
$$

where inequality (i) applies the triangle inequality to the relation (27) between the perturbed and idealized operators; step (ii) follows by definition of the residual $r^{(t)}$ at round $t$; and step (iii) follows from the bound (28) on the Lipschitz coefficient of $\mathcal{T}$. Performing induction on this bound yields the stated claim.

**Proof of the bound (28):** It remains to bound the Lipschitz coefficient of the idealized operator $\mathcal{T}$. Since the composite function $F(z) := \sum_{j=1}^m f_j(z_j)$ is $\ell_*$-strongly convex and $L^*$-smooth, known results on reflected proximal operators [11, Theorems 1 and 2] imply that with the stepsize choice $s = 1/\sqrt{\ell_* L^*}$, the operator $\mathbf{refl}_{sF}$ satisfies the bound

$$
\|\mathbf{refl}_{sF}(z) - \mathbf{refl}_{sF}(z')\|_2 \leqslant \left(1 - \frac{2}{\sqrt{\kappa}+1}\right)\|z - z'\|_2 \qquad \text{for all } z, z' \in (\mathbf{R}^d)^m.
\tag{30}
$$

On the other hand, the reflected proximal operator $\mathbf{refl}_{I_E}$ for the indicator function $\mathbf{refl}_{I_E}$ is non-expansive, so that

$$
\|\mathbf{refl}_{I_E}(z) - \mathbf{refl}_{I_E}(z)\|_2 \leqslant \|z - z'\|_2 \qquad \text{for all } z, z' \in (\mathbf{R}^d)^m.
\tag{31}
$$

Applying the triangle inequality and using the definition (25) of the idealized operator $\mathcal{T}$, we find that

$$
\begin{aligned}
\|\mathcal{T}(z) - \mathcal{T}(z')\|_2 &\leqslant \|\mathbf{refl}_{sF}\big(\mathbf{refl}_{I_E}(z)\big) - \mathbf{refl}_{sF}\big(\mathbf{refl}_{I_E}(z')\big)\|_2 \\
&\overset{\text{(iv)}}{\leqslant} \left(1 - \frac{2}{\sqrt{\kappa}+1}\right)\|\mathbf{refl}_{I_E}(z) - \mathbf{refl}_{I_E}(z')\|_2 \\
&\overset{\text{(v)}}{\leqslant} \left(1 - \frac{2}{\sqrt{\kappa}+1}\right)\|z - z'\|_2,
\end{aligned}
$$

where step (iv) uses the contractivity (30) of the operator $\mathbf{refl}_{sF}$, and step (v) uses the non-expansiveness (31) of the operator $\mathbf{refl}_{I_E}$. This completes the proof of the bound (28).

### A.2.3 Proof of Corollary 1

By construction, the function $h_j$ is smooth with parameter $M := sL^* + 1$ and strongly convex with parameter $m := s\ell_* + 1$. Consequently, if we define the operator $H_j(u) := u - \alpha\nabla h_j(u)$, then by standard results on gradient methods for smooth-convex functions, the stepsize choice $\alpha = \frac{2}{M+m}$ ensures that the operator $H_j$ is contractive with parameter at least $\rho = 1 - \frac{m}{M}$. Thus, we have the bound

$$
\|u^{(e+1)} - u^*\|_2 \leqslant \rho^e\|u^{(1)} - u^*\|_2,
$$

where $u^* = \mathbf{prox}_{sf_j}(x_j^{(t)})$ is the optimum of the proximal subproblem. Unpacking the definitions of $(m, M)$ and recalling that $s = 1/\sqrt{\ell_* L^*}$, we have

$$
\frac{M}{m} = \frac{sL^* + 1}{s\ell_* + 1} = \frac{\sqrt{\frac{L^*}{\ell_*}} + 1}{\sqrt{\frac{\ell_*}{L^*}} + 1} \leqslant \sqrt{\kappa} + 1,
$$

and hence $\rho \leqslant 1 - \frac{1}{\sqrt{\kappa}+1}$, which establishes the claim.

### A.2.4 Proof of Theorem 2

Recalling the definition (17) of the regularized objective $F_\lambda$, note that it is related to the unregularized objective $F$ via the relation $F_\lambda(x) = F(x) + \frac{m\lambda}{2}\|x - x^{(1)}\|^2$, where $x^{(1)}$ is the given initialization. The proposed procedure is to compute an approximation to the quantity

$$x_\lambda^\star := \arg\min_{x \in \mathbf{R}^d} \underbrace{\left( \sum_{j=1}^m \left\{ f_j(x) + \frac{\lambda}{2}\|x - x^{(1)}\|^2 \right\} \right)}_{=:F_\lambda(x)}.$$

Now suppose that we have computed a vector $\widehat{x} \in \mathbf{R}^d$ satisfies $F_\lambda(\widehat{x}) - F_\lambda(x_\lambda^\star) \leqslant \varepsilon/2$. Letting $F^\star = F(x^\star)$ denote the optimal value of the original (unregularized) optimization problem, we have

$$F(\widehat{x}) - F^\star = \left\{ F(\widehat{x}) - F_\lambda(x_\lambda^\star) \right\} + \left\{ F_\lambda(x_\lambda^\star) - F(x^\star) \right\}. \tag{32}$$

By definition of $F_\lambda$, we have $F(\widehat{x}) \leqslant F_\lambda(\widehat{x})$. Moreover, again using the definition of $F_\lambda$, we have

$$F_\lambda(x_\lambda^\star) - F(x^\star) = F_\lambda(x_\lambda^\star) - F_\lambda(x^\star) + \frac{m\lambda}{2}\|x^\star - x^{(1)}\|^2$$
$$\leqslant \frac{m\lambda}{2}\|x^\star - x^{(1)}\|^2,$$

where the inequality follows since $x_\lambda^\star$ minimizes $F_\lambda$ by definition. Substituting these bounds into the initial decomposition (32), we find that

$$F(\widehat{x}) - F^\star \leqslant \left\{ F_\lambda(\widehat{x}) - F_\lambda(x_\lambda^\star) \right\} + \frac{m\lambda}{2}\|x^\star - x^{(1)}\|^2$$
$$\leqslant \frac{\varepsilon}{2} + \frac{\varepsilon}{2} = \varepsilon. \tag{33}$$

where the inequality follows since since $\widehat{x}$ is $(\varepsilon/2)$-cost-suboptimal for $F_\lambda$, and by our selection of $\lambda$. Thus to finish the proof, we simply need to check how many iterations it takes to compute an $(\varepsilon/2)$-cost-suboptimal point for $F_\lambda$.

Let us define the shorthand notation $\overline{L} := \sum_{j=1}^m L_j$ and $\kappa_\lambda := \frac{L^\star + \lambda}{\lambda}$. Since $F_\lambda$ is a sum of functions that are $\lambda$-strongly convex and $(L_j + \lambda)$-smooth, it follows that from initialization $x^{(1)}$, the FedSplit algorithm outputs iterates $x^{(t)}$ satisfying the bound

$$F_\lambda(x^{(t+1)}) - F_\lambda(x_\lambda^\star) \overset{\text{(i)}}{\leqslant} \frac{\overline{L} + m\lambda}{2}\|x^{(t+1)} - x_\lambda^\star\|^2$$
$$\overset{\text{(ii)}}{\leqslant} \frac{\overline{L} + m\lambda}{2} \left( 1 - \frac{2}{\sqrt{\kappa_\lambda} + 1} \right)^{2t} \frac{\|x^{(1)} - z_\lambda^\star\|^2}{m}. \tag{34}$$

In the above reasoning, inequality (i) is a consequence of the smoothness of the losses $f_j$ when regularized by $\lambda$, along with the first-order optimality condition for $x_\lambda^\star$; and bound (ii) then follows by squaring the guarantee of Theorem 1 with $b = 0$. By inverting the bound (34), we see that in order to achieve an $\varepsilon/2$-optimal solution, it suffices to take the number of iterations $t$ to be lower bounded as

$$t \geqslant \left\lceil \frac{\sqrt{\kappa_\lambda} + 1}{4} \log \left\{ \frac{(\overline{L} + \lambda m)\|x^{(1)} - z_\lambda^\star\|^2}{m} \right\} \right\rceil.$$

Evaluating this bound with the choice $\kappa_\lambda = 1 + L^\star/\lambda$ and recalling the bound (33) yields the claim of the theorem.

### A.3 Characterization of fixed points

In this section we give the two fixed point results for FedSGD and FedProx as stated in Section **??**.

### A.3.1 Proof of Proposition 1

We begin by characterizing the fixed points of the `FedSGD` algorithm. By definition, any limit point $(x_1^\star, \ldots, x_m^\star) \in (\mathbf{R}^d)^m$ must satisfy the fixed point relation

$$x_j^\star = \frac{1}{m} \sum_{j=1}^{m} G_j^e(x_j^\star), \qquad j = 1, 2, \ldots, m.$$

Thus, the limits $x_j^\star$ are common, and this gives part (a) of the claim. Expanding the iterated operator $G_j^e$ gives part (b).

### A.3.2 Proof of Proposition 2

We now characterize the fixed points of the `FedProx` algorithm. By definition, any limit point $(x_1^\star, \ldots, x_m^\star)$ satisfies

$$x_j^\star = \frac{1}{m} \sum_{j=1}^{m} \mathbf{prox}_{sf_j}(x_j^\star), \qquad j = 1, 2, \ldots, m. \tag{35}$$

Thus, the limits $x_j^\star$ are common, and this gives part (a) of the claim.

For any convex function, $f\colon \mathbf{R}^d \to \mathbf{R}$, the proximal operator satisfies

$$\mathbf{prox}_{sf}(v) = v - s\nabla M_{sf}(v), \quad \text{for all } s > 0 \text{ and } v \in \mathbf{R}^d.$$

Using this identity in display (35) yields part (b) of the claim.

## B  Details for simulation studies

All of the experiments were conducted on a 2.6 GHz Intel Core i7 processor, in `Python` 3.7.3. Our logistic regression experiments used `CVXPY`, convex programming [10] software that we used to implement the exact proximal operators.

### B.1  Results presented in Figure 1

For the simulation, we construct a least squares problem where for $j \in [m]$, the response vector $b_j \in \mathbf{R}^{n_j}$ obeys the linear model $b_j = A_j x_0 + v_j$, where $x_0 \in \mathbf{R}^d$ is the unknown parameter vector to be estimated, and the noise vectors $v_j$ are independently distributed as $v_j \overset{\text{ind.}}{\sim} \mathsf{N}\left(0, \sigma^2 I_{n_j}\right)$ for some $\sigma > 0$. For our experiments reported here, we constructed a random instance of such a problem with $m = 25$, $d = 100$, $n_j \equiv 500$ and $\sigma^2 = 0.25$. We generated the design matrices with i.i.d. entries of the form $(A_j)_{kl} \overset{\text{i.i.d.}}{\sim} \mathsf{N}(0, 1)$, for $k = 1, \ldots, n_j$ and $l = 1, \ldots, d$. The aspect ratios of $A_j$ satisfy $n_j > d$ for all $j$, thus by construction the matrices $A_j$ are full rank with probability 1.

### B.2  Results presented in Figure 2

#### B.2.1  Synthetic dataset

Here, we have design matrices $A_j \in \mathbf{R}^{n_j \times d}$ and label vectors $b_j \in \{1, -1\}^{n_j}$. We denote the rows of $A_j$ by $a_{ij} \in \mathbf{R}^d$ for $i = 1, \ldots, n_j$. The conditional probability of positive class label $b_{ij} = 1$ under unknown parameter vector $x_0$ is then

$$\mathbf{P}\{b_{ij} = 1\} = \frac{\mathrm{e}^{a_{ij}^\mathsf{T} x_0}}{1 + \mathrm{e}^{a_{ij}^\mathsf{T} x_0}}, \quad \text{for } i = 1, \ldots, n_j. \tag{36}$$

Given observations of this form, we solve the *logistic regression* problem, This problem is smooth and convex, and clearly a special case of the more general class of federated problems (1).

We construct random instances of logistic regression problems with the settings $d = 100, n_j \equiv 1000$ and $m = 10$. Hence, we have a total sample size of $n = 10000$. We draw $a_{ij} \overset{\text{i.i.d.}}{\sim} \mathsf{N}(0, I_d)$ for all $i, j$ and $x_0 \overset{\text{i.i.d.}}{\sim} \mathsf{N}(0, I_d)$. The binary labels then are constructed to follow the Bernoulli model (36).

### B.2.2 FEMNIST datset

For this experiment only, we used Amazon EC2 to carry out these experiments (on `c5.metal` instances). The original dataset is comprised of $28 \times 28$ images, which we vectorize in row major order to obtain data points in $u_{ij} \in \mathbf{R}^{784}$. We further preprocessed these datapoints by adding a constant feature, and adding $(Ru)_+$ and $(Gu)_+$, where $R \in \{\pm 1\}^{3000 \times 784}$ and $G \in \mathbf{R}^{3000 \times 784}$ are filled with i.i.d. Rademacher and standard Normal entries. Here, $(\cdot)_+$ denotes the entrywise positive part of a vector. Therefore our final datapoints are

$$a_{ij} = (1, u_{ij}, (Ru_{ij})_+, (Gu_{ij})_+) \in \mathbf{R}^{6785}.$$

There were $K = 62$ classes in the dataset; we encode the labels as vectors $b_{ij} \in \{\pm 1\}^K$. Formally, if $a_{ij}$ belongs to class $k \in [K]$, we set $b_{ij} = 2e_k - \mathbf{1}$, where $e_k$ denotes the $k$th standard basis vector in $\mathbf{R}^K$.

We added the additional random features given above to improve the performance of our model on held out data. We set $\lambda = 0.01$ by cross-validation on a smaller subsample of the FEMNIST dataset. Formally, for each client, we select a random, $20\%$ fraction of the data to reserve as a heldout set, not used for training our classifier. We train the one-versus-all multiclass classifier, according to the objective given in (19) by `FedSplit`until approximately satisfying the optimality condition of the distributed problem. We then compute the accuracy of our multiclass classifier on the held out data and repeated this for choices of $\lambda \in [10^{-3}, 10^3]$; $\lambda = 0.01$ worked best on the held out data, giving an accuracy of $73\%$. As mentioned in the paper, the proximal solves for `FedSplit`were carried out using accelerated gradient descent.

### B.3 Results presented in Figure 3

We now describe the results of a simulation study that demonstrates the accuracy of these predicted iteration complexities. At a high level, our strategy is to construct a sequence of problems, indexed by an increasing sequence of condition numbers $\kappa$, and to estimate the number of iterations required to achieve a given tolerance $\varepsilon > 0$ as a function of $\kappa$. In order to do, it suffices to consider ensembles of least squares problems (8), but with a carefully constructed collection of design matrices, which we now describe.

For a given integer $\ell \geqslant 2$, let $\mathrm{O}(\ell)$ denote the set of $\ell \times \ell$ orthogonal matrices over the reals, and let $\mathsf{Unif}(\mathrm{O}(\ell))$ denote the uniform (Haar) measure on this compact group. With this notation, we begin by sampling i.i.d.random matrices

$$U_j^{(\kappa)} \sim \mathsf{Unif}(\mathrm{O}(n_j)) \quad \text{and} \quad V_j^{(\kappa)} \sim \mathsf{Unif}(\mathrm{O}(d)), \qquad \text{for } j = 1, \ldots, m. \tag{37}$$

For a given condition number $\kappa \geqslant 1$, we define a padded diagonal matrix—that is

$$\Lambda_j^{(\kappa)} = \begin{bmatrix} \mathbf{diag}(\lambda_j^{(\kappa)}) & 0_{d,(n-d)} \end{bmatrix} \quad \text{where} \quad \lambda_j^{(\kappa)} = (\sqrt{\kappa}, 1, \ldots, 1) \in \mathbf{R}^d.$$

Above, the matrix $0_{d,(n_j-d)} \in \mathbf{R}^{d \times (n_j-d)}$ has all entries equal to zero. Given the random orthogonal matrices and the matrix $\Lambda_j^{(\kappa)} \in \mathbf{R}^{n_j \times d}$, we then construct the design matrices $A_j^{(\kappa)} \in \mathbf{R}^{n_j \times d}$ by setting

$$A_j^{(\kappa)} := U_j^{(\kappa)} \Lambda_j^{(\kappa)} V_j^{(\kappa)}, \quad \text{for all } j = 1, \ldots, m.$$

These choices ensure that the federated least squares objective (8) has condition number $\kappa$.

As before, the response vectors $b_j^{(\kappa)}$ obey a Gaussian linear measurement model,

$$b_j^{(\kappa)} = A_j^{(\kappa)} x_0 + v_j^{(\kappa)}, \quad \text{for } j = 1, \ldots, m, \quad \text{and for all } \kappa \in K.$$

We again take $v_j^{(\kappa)} \overset{\text{ind.}}{\sim} \mathsf{N}\left(0, \sigma^2 I_{n_j}\right)$. In our experiments, we draw the parameter $x_0 \sim \mathsf{N}\left(0, I_d\right)$, and use the parameter settings

$$m = 10, \quad d = 100, \quad n_j \equiv 400, \quad \text{and} \quad \sigma^2 = 1.$$

With these settings, we iterated over a collection of condition numbers $\kappa \in \{10^0, 10^{0.5}, \ldots, 10^{3.5}, 10^4\}$. For each choice of $\kappa$, after generating a random instance as described above, we measured the number of iterations required for `FedGD` and the `FedSplit` procedures, respectively, to reach a target accuracy $\varepsilon = 10^{-3}$, which is modest at best.