[Reviews · NeurIPS 2020]

Review 1

Summary and Contributions: The paper presents a new splitting method for federated optimization. Convergence rates are established under a variety of plausible assumptions. These range for geometric rate under Lipschitz + strong convexity assumptions, to O(1/t^2) rates for smooth but not strongly convex functions. In all cases, exact (i.e., non-noisy gradients are considered).

Strengths: The results are new and the problem formulation is natural and useful. The method is elegant. Overall, I think the paper makes a good contribution to the literature, especially compared to many previous works whose results are much messier.

Weaknesses: - The main weakness is the lack of a good theoretical result in the case when the proximal steps are not solved exactly. In that case, Corollary 1 gives a bound for b, but this is given in terms of the iterates themselves. I suppose one can always try to assume that the optimization is done over a set of finite diameter, in which case the right-hand side can be naturally bounded; but in the absence of this assumption, the analysis is not fully complete. - The results are quite easy to obtain and the techniques come from a natural modification of well-known arguments. However, this might be just because the method is cleverly chosen so that the ultimate analysis is simple.

Correctness: I have not found any errors.

Clarity: The paper is well written.

Relation to Prior Work: This is adequately discussed, though I'm not super familiar with the existing literature.

Reproducibility: Yes

Additional Feedback: **Edit:** I have now read the author response. I do not think the authors do a good job of responding to my comments. Overall, my opinion of the opinion is unchanged and I'm leaving the rating at a 7. First, my complaint was about the inherently circular nature of the results: Theorem 1 bounds what happens to the iterates in terms of b, and then Corollary 1 bounds b in terms of iterates. Even if, as the authors reply, their analysis could handle the case when b changes from step to step, the conceptual problem remains. Typically, this is fixed either by adding a projection, or adding a separate argument that the iterates remain bounded even without a projection. Second, the authors list some technical contributions they make in terms of operator splitting, but frankly these do not sound very impressive. Overall, I like this paper because it is simple and cute, and I would like to see it published. However, it is not without weaknesses.


Review 2

Summary and Contributions: The authors showed that the fixed points of FedSGD and FedProx need not correspond to stationary points of the original optimization problem. Then they proposed FedSplit framework to deal with this issue.

Strengths: - Strong theory, in fact the authors derived an explicite formula for the fixed points of FedSGD and FedProx. -They gave example of situations where these points do not correspond to stationary points of the original optimization problem with particular focus on least squares. -They proposed FedSplit to handle this issue and they proved its convergence to the right points.

Weaknesses: -The experiments are not strong. -A numerical illustration is given only based on simple logistic regression and using FEMNIST. -The authors can strength the numerical part using bigger models and larger datasets.

Correctness: -In the proof of thm2 I can not see where the condition on the step size is used. -In the same Theorem, I am surprised about the condition on the step size. In fact the proposed step size is proportional to 1/sqrt(\lambda) and \lambda is of order of \epsilon thus the step size is of order of 1/sqrt(\epsilon)!! too big.

Clarity: The paper is clear and well written.

Relation to Prior Work: yes

Reproducibility: Yes

Additional Feedback: -See above my concern with thm2. -In equation (17) you used x^1 in the regularization term. It seems that other vectors can work as well? -I think explanation and comment on this need to be included in the paper. -Some experiments about the impact of this regularization are needed. ----After the rebuttal----- I read the other reviews and the author response. I especially understand from the other reviews that there is some criticism about the theoretical properties and their comparison with some results from the literature. Nevertheless, I still continue to think that this is a good submission, so I decided to keep my score unchanged to 7.


Review 3

Summary and Contributions: Main merit: The paper introduces a novel local algorithm for federated learning -- FedSplit. While the classical FL methods such as FedAvg and FedProx does not have a correct fixed point, FedSplit has the correct fixed points despite a quite simple structure. I believe this is a very nice contribution and should be appreciated by the community. _______________ After the rebuttal _________________ The authors did not disagree with the points I raised. At the same time, the authors have proposed a reasonable fixes to these issues: mention SCAFFOLD, be a little more careful about taking a credit for noticing wrong fixed points of FedAvg/FedProx and mention that rate of FedSplit is no better than rate of AGD in heterogenous setting (but argue the benefits of FedSplit in the iid setup). Just one last comment: I was not persuaded by the comment that FedProx is better to AGD in terms of the inexactness; I believe more details would need to be given to make this argument solid (I understand there is no space for this in the rebuttal). 1) if one has access to the exact gradients, AGD can be performed directly, while FedSplit still needs to solve the local subproblem inexactly, and 2) AGD can still work well under inexact updates one done correctly, see https://arxiv.org/abs/1109.2415 for example (one should make more detailed argument about inexactness over there and Thm 1,3 from your paper). Given the above (I read the other reviews too), I stand by my initial score -- 6.

Strengths: Explained in the contributions.

Weaknesses: Main criticism:  1) The paper claims two main contributions, one of which is  "The first contribution of this paper is to analyze some past procedures, and show that even in the favorable setting of deterministic updates (i.e., no stochastic approximation used), these methods typically fail to preserve solutions of the original optimization problem as fixed points " I believe the text above is misleading. In fact, it was already well known for the "past procedures" to not have the correct fixed points; one alternative approach to deal with such an issue was to incorporate the "drift"; see https://arxiv.org/abs/1910.06378 for example. Therefore, believe it would be more appropriate to not claim the contribution for showing the wrong fixed point of the local algorithms.  2) While FedSplit is an honest local algorithm with a correct fixed point; the convergence properties of FedSplit are strictly worse than those of the plain accelerated gradient descent (AGD). Specifically, in strongly convex case (same arguments apply for weakly convex), the communication complexity of FedSplit is O(sqrt(kappa)log(1/epsilon)), which is identical to the communication complexity of AGD. In fact, AGD is favorable (in terms of the rate) as is requires a single gradient evaluation instead of evaluating the prox with high enough precision so that the inexactness does not drive the rate.  To be fair, the local methods do not, in general, outperform the non-local counterparts in terms of the communication even if the fixed point is correct; see https://arxiv.org/abs/2006.04735 for example (no need to cite it as the paper appeared online only recently) -- in fact, one might argue that plain AGD is optimal in such a case (see https://arxiv.org/abs/2005.10675 for example). For that reason, one might not hope for a better rate of FedSplit.  To "fix" this issue, I suggest the following: i) consider data homogeneous setting (i.e., identical data across nodes). In such a case, FedSplit should strictly outperform AGD (I presume it would even converge in a single iteration in such a case) ii) mention transparently in the paper that FedSplit is not favorable over AGD in terms of the theory for the heterogenous setting.

Correctness: I did a high-level check of the proofs and all seem correct.

Clarity: I am happy with the level of writing of the paper (besides the )

Relation to Prior Work: Yes, the difference and motivation of this work is well justified.

Reproducibility: Yes

Additional Feedback: See Weaknesses


Review 4

Summary and Contributions: The paper considers federated distributed optimization under convex assumption. The paper first identifies the fixed points of some federated optimization algorithms may not be stationary points of the original optimization. To fix this, the paper introduces a new algorithm which has fixed points corresponding to the optima of the original problem. The convergence rate is also established for the proposed algorithm.

Strengths: The paper presents the federated SGD and federated proximal algorithms have fixed points not corresponding to the zero of sum of gradients of the consensus problems even in the deterministic case. An example is presented to verify the claim. The paper then proposes the FedSplit algorithm based on operator splitting algorithm with additive structure. The algorithm has a local proximal update local parameters and aggregates local parameters to update the server. The common convergence rate is established for the algorithm using common tools.

Weaknesses: The paper could add more details on the derivation of the algorithm 1 and give some intuition about why the proposed algorithm could fix the issue. The FedSplit method is more like deterministic distributed optimization algorithm. The connections to the multi-device communications and failures mentioned in the paper are weak.

Correctness: The algorithm looks correct and convergence rates are classical optimization results using classical tools.

Clarity: The paper is well written. It presents good motivation and example, followed by the proposal to address the issue. The experiment is designed to show the advantages of the new method.

Relation to Prior Work: The connections to the existing work are clearly stated, and the contributions are also clearly presented.

Reproducibility: Yes

Additional Feedback:

[Author Response · NeurIPS 2020]

We thank the reviewers for their careful reading and feedback. We now address their comments in turn.

**Reviewer 1:**  *"The main weakness is the lack of a theoretical result . . . when proximal steps are not solved exactly."*
To clarify, our submission does contain more general inexact convergence results. Specifically, Theorem 3 (Appendix A.2.1, pp. 13)
contains results when the residuals for the proximal updates are allowed to vary, and are not uniform across iterations. Our revision
includes a mention of these more general results following the statement of Theorem 1 in the main text. Additionally, from the proof
of Theorem 3, it is easy to see that if the inexactness conditions (23) (respectively, (13)) hold only in expectation (for example, if one
uses a stochastic gradient method to implement the proximal updates) then the conclusions (24) (respectively, (14)) also hold in
expectation. We will mention this extension following the presentation of Theorem 3. These results should allow a practitioner to
translate standard results regarding iteration complexity of (stochastic) gradient methods to this setting.

*"The results are quite easy to obtain . . . "*
While we view the simplicity of the argument as a positive feature, it is worthwhile emphasizing two new technical contributions in
terms of the theory of operator splitting methods for distributed convex optimization. The first is the inexact updates (Theorems 1
and 3): we agree that the corresponding results are known for gradient methods, but in the context of Peaceman-Rachford and similar
splitting procedures, Theorem 3 (Appendix A.2.1, pp. 13) is to our knowledge new. The second technical contribution to highlight is
our analysis of non-strongly convex problems via reduction to weakly convex problems. Although well known for gradient methods,
the $\tilde{O}(1/t^2)$ rate for non-strongly convex problems that we give in Theorem 2 is to our knowledge new for Peaceman-Rachford. Our
revision will mention this following the statements of Theorems 2 and 3.

**Reviewer 2:**  *"In the proof of Theorem 2, I cannot see where the condition on the step-size is used . . . I am surprised about the*
*condition on the step-size . . . It seems that [vectors other than $x^{(1)}$] can be used. Some explanation needs to be included . . . "*
To clarify, the condition on the stepsize is used in ll. 415 "by squaring the guarantee of Theorem 1. . . " in the proof of Theorem 2.
Specifically, after regularizing, the losses become $\lambda$-strongly convex and $(L_j + \lambda)$-smooth, and so the correct stepsize scales in the
inverse of the geometric mean of these parameters (see Theorem 1). Although one can regularize around other points in the reduction
we employ, inequality (34) will be less clean, and the iteration complexity (see ll. 417) will be less clean, in that it would depend on
more than the initial conditions (distance of the initializer to opt). Although the inverse dependence on epsilon (note that our step
size is $1/\sqrt{\varepsilon^2 + L^*\varepsilon}$) may at first seem counter-intuitive, note that when $\varepsilon$ is large, this means the weight of the objective is small
as compared to the effect of the regularization term. Moreover, this type of scaling has been noted previously for gradient-based
reductions from weak to strong convexity [3].

*"The experiments are not strong . . . [use] bigger models and larger datasets."*
The extended version of our paper (on arXiv) includes additional experiments on larger datasets, that show our procedure's behavior
on easy and difficult problem instances, as measured by problem conditioning. We will include a citation in the experiments section
that points to these simulations.

**Reviewer 3:**  *". . . the convergence properties of FedSplit are strictly worse than those of AGD . . . mention transparently in the*
*paper . . . in [the homogeneous setting], FedSplit should strictly outperform AGD.""*
We agree that the iteration complexity of FedSplit and AGD are comparable, assuming exact gradient computations/exact proximal
evaluations and in this setting one should prefer AGD. On the other hand, it is known that AGD is sensitive to inexact updates
(see [1]). In comparison, our inexact convergence guarantees show that FedSplit is robust to noisy updates (see Theorems 1 and 3).
We also agree that if the proximal stepsize is sufficiently small (as compared to the desired accuracy level), then FedSplit dominates
AGD in iteration complexity in the homogeneous setting. Our revision mentions these points in the paragraph preceding section 5.

*"In fact, it was already well known for the 'past procedures' to not have the correct fixed points."*
We thank the reviewer for bringing SCAFFOLD [2] to our attention. We were not familiar with this work prior to this submission.
While there is obvious conceptual overlap (and we will of course cite it in the revision), based on our reading, this paper does not
explicitly demonstrate that FedProx or FedAvg have incorrect fixed points. To further clarify, we have rephrased this contribution as
"We demonstrate that procedures such as FedProx and FedGD do not generally have correct fixed points, even for simple quadratic
objectives." We believe this is a more specific and accurate characterization of our results.

**Reviewer 4:**  *"The `FedSplit` method is more like a deterministic distributed optimization algorithm. The connections to the*
*multi-device communications and failures mentioned in the paper are weak."*
Although operator splitting has been used successfully in standard distributed optimization settings, we believe that the mild
dependence on condition number (which in the context of federated optimization corresponds to reduced rounds of communication)
is important in multi-device communication. Additionally, our inexact guarantees as stated in Theorems 1 and 3 are especially
important in multiparty communication occurring during federated optimization: device updates may be inexact due to computational
constraints or errors may arise during communication. We have made these connections clearer following the statement of Theorem
1 in our revision. Finally, we agree that device failures are not addressed in the present work; we have removed the mention of this
topic in the broader impact statement. (Our ongoing work has obtained results in this setting.)

*"The paper could add more details on the derivation of algorithm 1 and give some intuition. . . "*
Thank you for this feedback. We agree that additional explanation is warranted here. Our revision now includes an additional
paragraph in section 3.1, explaining the relationship between problem (2), its optimality conditions, and a monotone inclusion
problem that results with Algorithm 1. This additional background should help readers less familiar with operator splitting techniques.

[1] O. Devolder, F. Glineur, and Y. Nesterov. First-order methods of smooth convex optimization with inexact oracle. *Mathematical Programming*, 146(1–2):37–75,
2014.

[2] S. P. Karimireddy, S. Kale, M. Mohri, S. J. Reddi, S. U. Stich, and A. T. Suresh. SCAFFOLD: stochastic controlled averaging for on-device federated learning.
Technical Report arxiv.org/abs/1910.06378, October 2019.

[3] Z. Allen Zhu and E. Hazan. Optimal black-box reductions between optimization objectives. In *Advances in Neural Information Processing Systems 29: Annual*
*Conference on Neural Information Processing Systems 2016, December 5-10, 2016, Barcelona, Spain*, pages 1606–1614, 2016.


[Meta-Review · NeurIPS 2020]

The paper first gives examples of fixed points for FedAvg/FedProx which are not corresponding to the zero of sum of gradients of the consensus problems even in the deterministic case. Motivated by the incorrect fixed points of FedAvg/FedProx, the paper then proposes FedSplit, a splitting method for federated optimization with convergence analysis. Consensus was reached among the reviewers that the contribution is valuable and above the bar for NeurIPS. However, we urge the authors to incorporate a discussion of e.g. Scaffold which solves largely the same problem, so that also the convergence rates should be compared, as well as more discussion AGD and on deterministic vs stochastic modes. We hope the detailed feedback with improvement suggestions from the 4 reviews will be implemented for the camera ready version.